# Plastoquinone homoeostasis by *Arabidopsis* proton gradient regulation 6 is essential for photosynthetic efficiency

Thibaut Pralon [1], Venkatasalam Shanmugabalaji [1], Paolo Longoni [1], Gaetan Glauser[1,2], Brigitte Ksas[3], Joy Collombat[1], Saskia Desmeules[1], Michel Havaux[3], Giovanni Finazzi [4] & Felix Kessler [1]

Photosynthesis produces organic carbon via a light-driven electron flow from $H_2O$ to $CO_2$ that passes through a pool of plastoquinone molecules. These molecules are either present in the photosynthetic thylakoid membranes, participating in photochemistry (photoactive pool), or stored (non-photoactive pool) in thylakoid-attached lipid droplets, the plastoglobules. The photoactive pool acts also as a signal of photosynthetic activity allowing the adaptation to changes in light condition. Here we show that, in *Arabidopsis thaliana*, proton gradient regulation 6 (PGR6), a predicted atypical kinase located at plastoglobules, is required for plastoquinone homoeostasis, i.e. to maintain the photoactive plastoquinone pool. In a *pgr6* mutant, the photoactive pool is depleted and becomes limiting under high light, affecting short-term acclimation and photosynthetic efficiency. In the long term, *pgr6* seedlings fail to adapt to high light and develop a conditional variegated leaf phenotype. Therefore, PGR6 activity, by regulating plastoquinone homoeostasis, is required to cope with high light.

[1] Faculty of Sciences, Laboratory of Plant Physiology, University of Neuchâtel, 2000 Neuchâtel, Switzerland. [2] Faculty of Sciences, Chemical Analytical Service of the Swiss Plant Science Web, Neuchâtel Platform for Analytical Chemistry (NPAC), University of Neuchâtel, 2000 Neuchâtel, Switzerland. [3] Commissariat à l'Energie Atomique et aux Energies Alternatives (CEA), Cadarache, Centre National de la Recherche Scientifique (CNRS), UMR 7265, Institut de Biosciences et de Biotechnologies d'Aix-Marseille, Laboratoire d'Ecophysiologie Moléculaire des Plantes Aix Marseille Université, 13108 Saint-Paul-lez-Durance, France. [4] Laboratoire de Physiologie Cellulaire et Végétale, UMR 5168, Centre National de la Recherche Scientifique (CNRS), Commissariat à l'Energie Atomique et aux Energies Alternatives (CEA), Institut National de la Recherche Agronomique (INRA), Institut de Biosciences et Biotechnologie de Grenoble (BIG), CEA-Grenoble Université Grenoble Alpes (UGA), 38000 Grenoble, France. Correspondence and requests for materials should be addressed to P.L. (email: paolo.longoni@unine.ch) or to F.K. (email: felix.kessler@unine.ch)

Oxygenic photosynthesis exploits light energy to generate an electron flow from $H_2O$ to NADPH, which is used for the production of organic molecules from $CO_2$. This process requires the coordinated activity of several membrane embedded complexes: photosystem II (PSII), the cytochrome $b_6f$ and photosystem I (PSI), which are functionally connected by diffusible electron carriers[1,2]. A membrane-soluble prenyl quinone, plastoquinone (PQ), ensures the electron transport between PSII and cytochrome $b_6f$[1–3]. High light intensities generate stress at the photosynthetic apparatus with PSII being particularly exposed to damage. By transferring electrons to cytochrome $b_6f$ PQ releases the light excitation pressure on PSII and as a consequence, prevents light-induced damage on the photosynthetic apparatus. However, only part of the total PQ participates in electron flow. This portion (the photoactive PQ pool) can be quantified by measuring its reduction by PSII[4,5] or its oxidation by PSI[6] via the cytochrome $b_6f$ complex upon light exposure. The remaining portion of the total PQ is not directly involved in photochemistry. This is defined as the non-photoactive PQ pool since it cannot be reduced by PSII or oxidised by PSI. This second pool of PQ is largely stored in lipid droplets associated with the thylakoid membranes: the plastoglobules[7]. The non-photoactive pool is involved in biosynthetic pathways occurring within the chloroplast (e.g. plastochromanol-8 biosynthesis[8]) and at the same time acts as an indispensable reservoir of PQ to refill the photoactive pool. In fact, when a plant experiences light intensities exceeding its electron transport capacity (high light), the photoactive PQ pool is damaged[9,10]. By replenishing it, the presence of a sufficient non-photoactive PQ pool reservoir ensures photosynthetic efficiency under prolonged stressful light conditions[4,11,12].

The proton gradient regulation (PGR) family comprises mutants displaying a perturbation of photosynthetic electron transport[13], which in turn compromises the formation of a proton gradient across the thylakoid membranes. The proton gradient not only aliments ATP synthesis but also induces non-photochemical quenching (NPQ) of chlorophyll fluorescence upon high light exposure. *PGR6* codes for a predicted atypical **a**ctivity of **b**c1 **c**omplex **k**inase 1 that is localised inside the chloroplast and associated with plastoglobules[14–16]. The *pgr6* mutant is defective in NPQ and maximal photosynthetic electron transport rate[13,17]. Moreover, loss of PGR6 leads to developmental defects such as impaired cotyledon greening and hypocotyl elongation under pure red light, which were reported to be independent from phytochrome-dependent light signalling pathways[18]. Upon several days of high light exposure, the *pgr6* mutant is characterised by growth and specific metabolic defects, such as low carotenoid accumulation and impaired sugar metabolism, which have been reported for adult plants[17,19].

In this study, we show that the *pgr6* primary defect consists in the misregulation of the homoeostatic relationship between the photoactive PQ pool and the non-photoactive PQ pool. By relating photophysiological measurements to the analysis of photochemically active and non-active PQ pools in wild type, *pgr6* and a mutant of PQ biosynthesis, we conclude that PGR6 is required to maintain the balance between the two pools already during a short (3 h) exposure to high light. This primary *pgr6* phenotype brings on the downstream defects in chloroplast physiology and plant development, which result in leaf variegation in high light exposed seedlings.

## Results

**Short-term photosynthetic defects in *pgr6*.** Phenotypic observation of young seedlings grown under continuous high light (500 µmol m$^{-2}$ s$^{-1}$) revealed that the *pgr6* mutation resulted in a chloroplast developmental issue that became visible as a conditional variegation of young leaves (Fig. 1a). Conversely, the same mutant plants did not show any visible phenotype when grown under continuous low light intensity (80 µmol m$^{-2}$ s$^{-1}$). This variegation is reminiscent of the phenotype previously reported in plants affected in protein turnover[20], reoxidation of PQ[21] or chloroplast to nucleus signalling[22]. Thus, this observation suggests that PGR6 is part of a mechanism essential for chloroplast development under high light.

To pinpoint the *pgr6* primary defect, while limiting the secondary effects that may arise from this initial perturbation[17,19], we grew wild type and mutant plants under moderate light conditions (120 µmol m$^{-2}$ s$^{-1}$, 8 h light/16 h dark) for 5 weeks and then exposed them to a relatively short high light treatment (3 h, 500 µmol m$^{-2}$ s$^{-1}$). This did not cause lasting damage to the photosynthetic apparatus, as shown by the maximum PSII efficiency ($\Phi_{MAX} = F_V/F_M$), which remained similar in both *pgr6* lines and wild-type plants (Fig. 1b). However, both the PSII quantum yield ($\Phi_{PSII}$) and the NPQ were clearly lower in both *pgr6* mutant lines compared to wild type (Fig. 1c, d). The photosynthetic defects observed in *pgr6* after 3-h high light were not due to alterations in the composition and/or abundance of photosynthetic complexes that were present at comparable levels of representative subunits of PSII (D1 (PsbA) and PsbO), PSI (PsaD and PsaC), light harvesting complex (LHCII) (Lhcb2), cytochrome $b_6f$ (cytb6f) (PetC) and the ATPase (AtpC) (Supplementary Fig. 1).

**Loss of PGR6 affects state transition kinases activity.** When shifted to high light plants acclimate via changes in the phosphorylation patterns of their photosynthetic protein complexes. High light leads to the inactivation of state transition kinase 7 (STN7) that phosphorylates the PSII antenna and a concomitant increase in the phosphorylation of the PSII core proteins by state transition kinase 8 (STN8)[23]. Since PGR6 is a predicted atypical kinase that may phosphorylate chloroplast proteins[14–17], we investigated whether the observed modifications in the $\Phi_{PSII}$ and NPQ parameters reflect a modification in the phosphorylation of the photosynthetic complexes. We analysed the phosphorylation patterns of major thylakoid proteins in wild type, *pgr6-1* and *pgr6-2* by anti-phosphothreonine immunoblotting and discovered that phosphorylation of both LHCII and PSII was clearly lower in both *pgr6* lines after 3-h high light compared to wild type (Fig. 2a), while there was no visible difference under moderate light. We found that the phosphorylation of the two major LHCII subunits (Lhcb1 and Lhcb2), as assessed by phosphorylation-dependent band shift using Phostag™-gels, was severely decreased in *pgr6* upon high light exposure. This result suggests that in *pgr6*, the activity of the STN7 kinase is more severely downregulated compared to wild type (Fig. 2b). Indeed, STN7 is the principal responsible for the phosphorylation of the trimeric LHCII, triggering the migration of mobile LHCII trimers between the two PSs in a process known as state transitions[24]. STN7 activity depends on the reduction of photoactive PQ at cyt$b_6f$,[25–27] and PGR6 is associated with plastoglobules (thylakoid-attached lipid droplets) that are believed to function as PQ reservoir[7,14–16]. We therefore reasoned that the observed changes in the phosphorylation patterns might reflect a perturbation in PQ redox state and/or availability in this mutant rather than a direct effect of PGR6 kinase activity.

We thus investigated the impact of the *pgr6* mutation on state transitions. To follow the antenna movement in vivo, we measured the chlorophyll fluorescence at room temperature in plants while switching from red supplemented with far-red (FAR) light (which triggers LHCII dephosphorylation leading to state 1) to red light only (which enhances phosphorylation in state 2)[28].

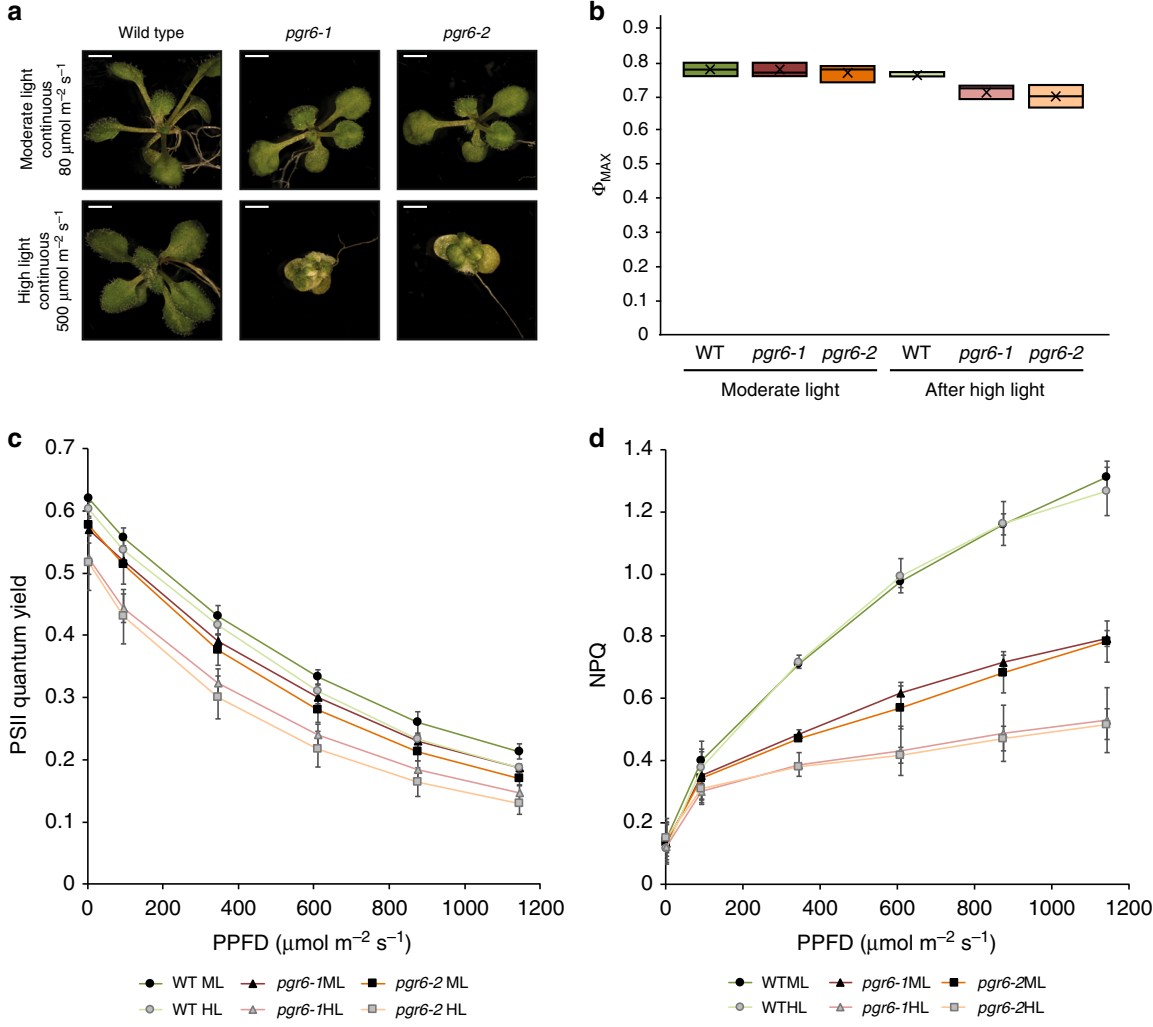

**Fig. 1** The *pgr6* mutant has a conditional variegated phenotype and is affected in photosystem II efficiency. **a** Visible phenotype of 14 days seedlings of wild type, *pgr6–1* and *pgr6-2* mutants grown on 0.5 × MS medium under 80 μmol m$^{-2}$ s$^{-1}$ and 500 μmol m$^{-2}$ s$^{-1}$ in 24 h continuous light; scale bar = 2 mm. Twenty-four-day-old plants grown on soil in short day cycle (8 h light /16 h dark) were used to assess the photosynthetic efficiency of wild type (WT), *pgr6-1* and *pgr6−2* under moderate light (120 μmol m$^{-2}$ s$^{-1}$) (ML, dark colours) and after 3 h of high light (500 μmol m$^{-2}$ s$^{-1}$) (HL, light colours). After 10 min of dark relaxation, variable room temperature chlorophyll fluorescence was measured on whole plants exposed to these light conditions to determine the following parameters: **b** PSII maximum quantum yield ($\Phi_{MAX} = F_V/F_M$). **c** PSII quantum yield ($\Phi_{PSII} = (F_M'-F_S)/F_M'$) at increasing light intensities, whiskers and box plot shows the minimum, first quartile, median, average, third quartile and maximum of each dataset. **d** Non-photochemical quenching (NPQ = $(F_M - F_M')/F_M'$) after 1 min of exposure at different light intensities. These measures were performed with a Fluorcam (MF800 – PSI) with blue light LEDs (470 nm). Each value represents the average of a pot containing 2–3 plants. Error bars indicate ± SD (*n* = 3 biologically independent samples). Data points for the items **b**–**d** are available in Supplementary data 1

Decrease in the fluorescence level upon light switch is a proxy for the antenna movement and is absent in the state transition mutant *stn7*[24]. Consistent with the phosphorylation data, we found that both *pgr6* and wild type have the same capacity to undergo the state 1 to state 2 transition, when grown under and exposed only to moderate light (Fig. 2c). However, the transition from state 1 to state 2 was inhibited in *pgr6* but not in wild type after 3-h high light (Fig. 2c, d), supporting the idea that the loss of PGR6 causes a conditional state transition defect, which may be observed only after high light exposure.

It has been reported that the activation of STN7 kinase involves its own phosphorylation[26,27]. Therefore, we analysed the phosphorylation pattern of the STN7 protein using Phostag™ gels, and found that it was less phosphorylated in *pgr6* after high light treatment (Fig. 2e), once again in agreement with a perturbation of state transitions (i.e. STN7 activity). Interestingly, the phosphorylation of the PSII reaction centre proteins (i.e. D1

(PsbA) and D2 (PsbD)), which mostly depends on STN8[24,29], was also affected in *pgr6*. Three-hour high light exposure resulted in a lower phosphorylation level of D1 (PsbA) and D2 (PsbD) in *pgr6* compared to wild type (Fig. 2a and Supplementary Fig. 2). Although the regulation of STN8 has not been fully clarified yet, this observation suggests that the activities of the two state transition kinases are linked and possibly both dependent on the status of the photosynthetic electron transport chain (ETC) or that they have overlapping target proteins[30]. A decrease in PSII phosphorylation may affect the repair cycle of the core protein D1 (PsbA) thereby decreasing the maximum efficiency of PSII[31]. This did not appear to be the underlying cause of the long-term *pgr6* phenotype, as short high light exposure did not cause a measurable decrease in the maximum yield of PSII (Fig. 1b) or increase of the basal level of fluorescence in the dark ($F_0$). Both parameters are dependent on PSII activity and are affected if there is a defect in its repair cycle[32] (Supplementary Fig. 2).

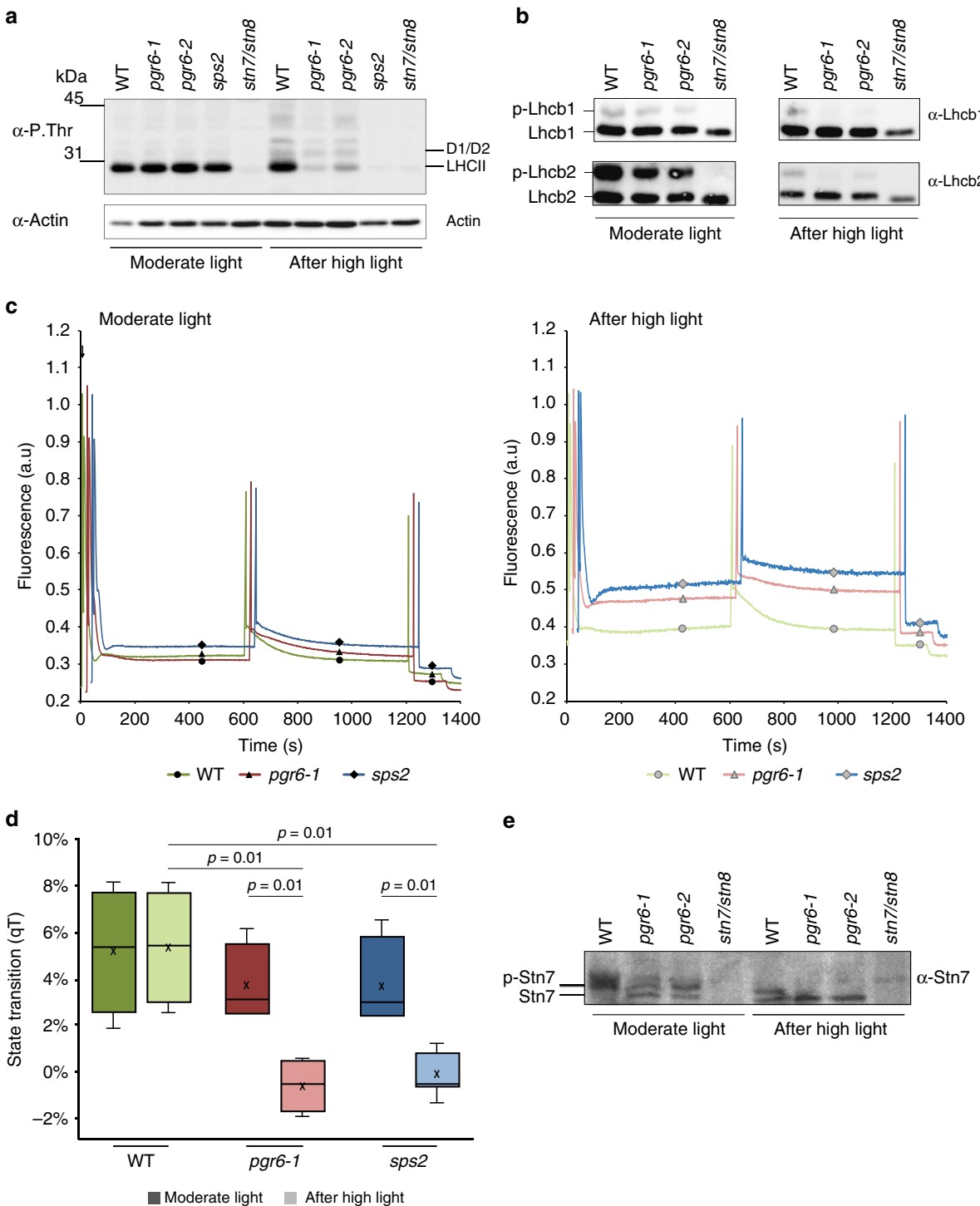

**Loss of PGR6 affects the photoactive PQ pool**. To address whether the defects in protein phosphorylation and state transitions are linked to a decrease in PQ availability, we compared the photosynthetic behaviour of *pgr6* to that of *sps2* (solanesyl phosphate synthase 2 mutant). In *sps2*, a mutant partially defective in PQ biosynthesis, the total PQ and, as a consequence, the photoactive PQ pools are decreased[4]. In particular, the shortage of photoactive PQ should diminish the electron capacity of the ETC affecting the photosynthetic efficiency. This would result in lower electron transport rate, and therefore a reduced quantum yield of the PSII ($\Phi_{PSII}$) and NPQ induction[4] (Supplementary Fig. 3). Since *sps2* by its nature is PQ-limited, it can be

used as a term of comparison to pinpoint a *pgr6* defect due to PQ availability. The first observation was that in *sps2*, as in *pgr6*, the thylakoid proteins are strongly dephosphorylated after 3-h high light (Fig. 2a). This result supports the hypothesis linking the downregulation of the STN7 and STN8 activity and perturbation of PQ availability. To directly assess the capacity of the ETC, we measured chlorophyll *a* fluorescence induction kinetics and calculated the electron transport capacity from the normalised area above the fluorescence traces[33] (Fig. 3a). The rationale for this choice is that the area above the fluorescence kinetics is a proxy of the average number of turnovers of each PSII reaction centre, i.e. of the number of electrons that this photosystem is able to inject

**Fig. 2** Thylakoid protein phosphorylation and state transitions are disturbed after high light treatment in *pgr6* background. **a** Total protein extracts of 4-week-old wild type (WT), *pgr6-1*, *pgr6−2*, *sps2* and *stn7/stn8* analysed by immunoblotting with anti-phosphothreonine antibody; the principal thylakoid phospho proteins are indicated on the right according to their size. Core photosystem II proteins D1 (PsbA) and D2 (PsbD) are indicated as a single band due to their poor resolution. Actin was used as a loading control. **b** Lhcb1 and Lhcb2 phosphorylation levels were visualised after separation on Phostag™-pendant acrylamide gels. The upper band corresponds to the phosphorylated form (p-), *stn7/stn8* double mutant is a non-phosphorylated control. **c** Average transient of the variable room temperature chlorophyll fluorescence measured during the transition from red (660nm) supplemented with far-red light (720nm) state 1 to pure red light state 2 ($n = 4$ independent pots containing 2–3 plants). The fluorescence curves from *pgr6* and *sps2* are shifted on the *x*-axis to allow visualising the $F_M$ST1 and $F_M$ST2 values. The *x*-axis time scale refers to the wild-type curve. **d** Calculated quenching related to state transition ($qT = (F_M ST1 – F_M ST2)/F_M$), expressed as the percentage of $F_M$ that is dissipated by the state 1 to state 2 transition, of wild type (WT), *pgr6−1* and *sps2* under moderate light ($120\mu mol\ m^{-2}\ s^{-1}$) (ML) and after 3 h of high light ($500\mu mol\ m^{-2}\ s^{-1}$) (HL). Whiskers and box plot shows the minimum, first quartile, median, average, third quartile and maximum of each dataset ($n = 4$ biologically independent samples); *p*-values are calculated via a two-tailed Student's *t* test. **e** STN7 phosphorylation level visualised after separation on Phostag™-pendant acrylamide gels. The upper band corresponds to the phosphorylated form (p-), a protein sample from *stn7/stn8* double mutant was loaded as a control for the antibody specificity. Uncropped images of the membranes displayed in **a**, **b** and **e** are available as Supplementary Fig. 11. Data points for items **c**, **d** are available as Supplementary data 2

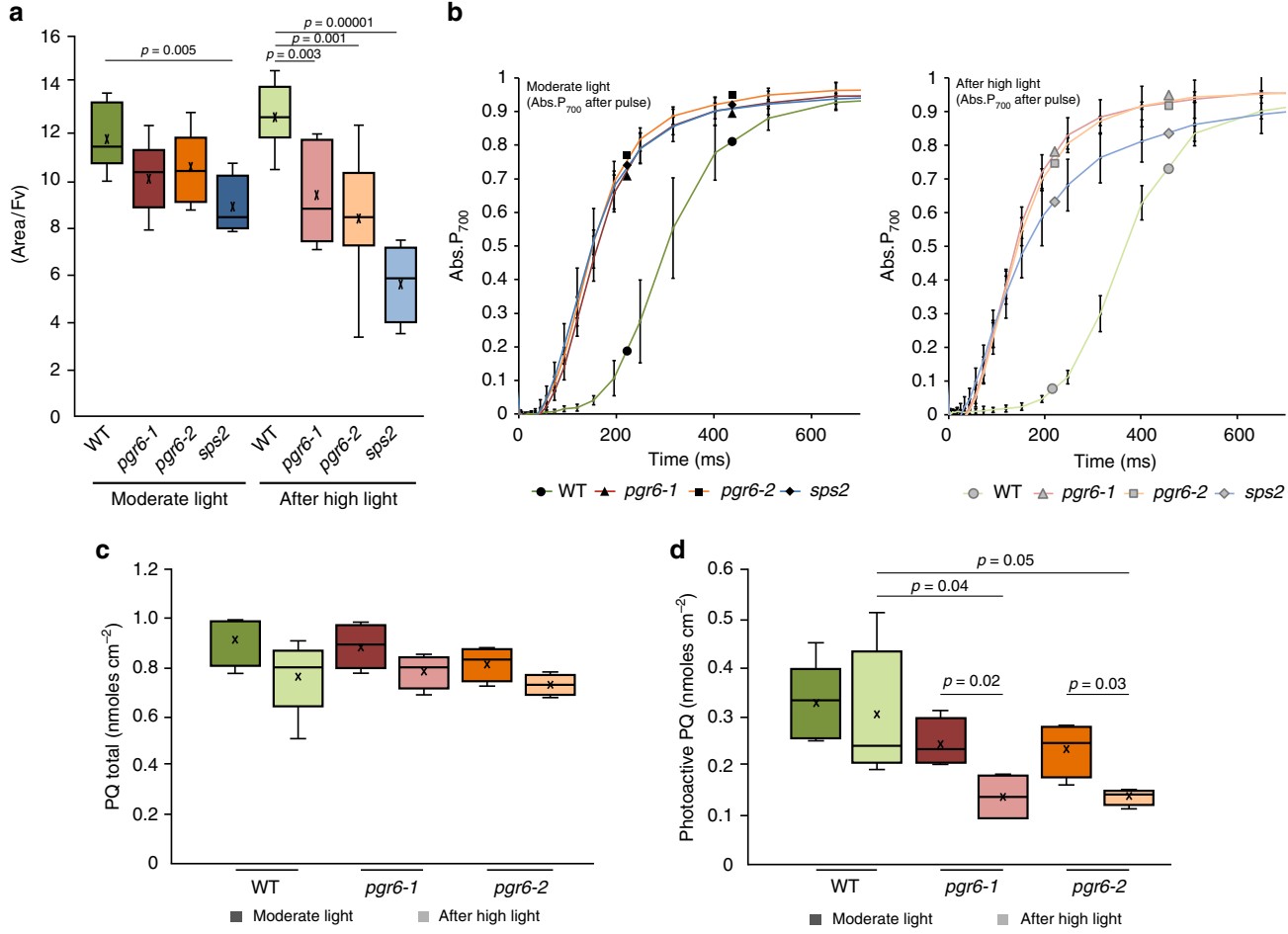

**Fig. 3** The *pgr6* mutant shows a limitation in photosynthetic electron carriers. Wild type (WT), *pgr6-1*, *pgr6−2* and *sps2* plants were grown under moderate light intensity and sampled under growth light conditions (moderate light) or after the exposure to 3 h at $500\mu mol\ m^{-2}\ s^{-1}$ (after high light). **a** Normalised area above the rapid fluorescence induction curve measured after 15 min of incubation in dark. This value estimates the number of available electron carriers per reaction centre (area/Fv). Error bars indicate ± SD ($n = 8$ for wild type, *pgr6−2*, $n = 6$ for *pgr6−1*, $n = 4$ for *sps2* biologically independent samples). **b** Analysis of $P_{700}$ re-oxidation kinetics induced by far-red light after a saturating light pulse (Time 0 = far-red ON). The oxidation status was measured by the increase in absorption at 810 nm on fully expanded leaves of each genotype. Error bars indicate ± SD ($n = 9$ biologically independent samples). **c** Leaf discs from wild type (WT), *pgr6-1* and *pgr6-2* plants were collected under moderate light ($120\mu mol\ m^{-2}\ s^{-1}$) and after 3 h of high light ($500\mu mol\ m^{-2}\ s^{-1}$), the average total plastoquinone (oxidised + reduced PQ) content was measured in nmoles $cm^{-2}$. **d** The oxidised and reduced PQ amount was measured in leaf discs exposed either to 2min far-red light ($5.5\mu mol\ m^{-2}\ s^{-1}$) to fully oxidise the photoactive PQ pool or to 15 s saturating flash ($2000\mu mol\ m^{-2}\ s^{-1}$) to fully reduce the photoactive PQ. The total photoactive PQ pool was calculated from the difference between these two conditions. Average values are reported as PQ nmoles $cm^{-2}$. Whiskers and box plot shows the minimum, first quartile, median, average, third quartile and maximum of each dataset ($n = 4$ biologically independent samples); *p*-values are calculated via a two-tailed Student's *t* test. Data points for item **a** are available as Supplementary data 3; data points for item **b** are available as Supplementary data 4; data points for items **c**, **d** are available as Supplementary data 6

**Table 1 The electron transport capacity per photosystem I is limited in *pgr6***

**Moderate light**

|  | Wild type | *pgr6-1* | *pgr6−2* | *sps2* |
|---|---|---|---|---|
| Lag time after far red (ms) | 8.78 ± 0.79 | 9.89 ± 1.06 | 8.57 ± 1.15 | 7.32 ± 1.06 |
| Lag time after pulse (ms) | 165.88 ± 2.93[a] | **74.34 ± 5.93[b]** | **69.38 ± 5.81[b]** | **64.54 ± 5.92[b]** |
| Calculated pulse e⁻ number | 19.35 ± 1.95[f] | **7.53 ± 0.41[g]** | **8.16 ± 0.68[g]** | **9.19 ± 2.23[g]** |

**After high light**

|  | Wild type | *pgr6-1* | *pgr6-2* | *sps2* |
|---|---|---|---|---|
| Lag time after far red (ms) | 17.14 ± 4.37 | 15.68 ± 2.94 | 15.01 ± 2.69 | 17.67 ± 0.98 |
| Lag time after pulse (ms) | 213.03 ± 12.68[c] | **54.04 ± 13.72[d]** | **47.36 ± 14.05[d,e]** | **14.75 ± 3.27[e]** |
| Calculated pulse e⁻ number | 13.21 ± 2.33[h] | **3.69 ± 1.51[i]** | **3.30 ± 1.31[i]** | **0.83 ± 0.16[i]** |

Electron (e⁻) capacity of the electron transport chain per photosystem I calculated from the $P_{700}$ oxidation lag time after a saturation pulse normalised over the lag time of the oxidation after dark. Values statistically different from the wild type for each condition are in bold. Superscript letters are used to indicate statistically different groups by Student's *t* test ($p < 0.05$) ($n = 4$ for wild type, *pgr6-1* and $n = 3$ for *pgr6−2*, *sps2* biologically independent samples). Data points are available in Supplementary data 4

into the ETC. It thus provides quantitative data on all the ETC electron acceptors, including the PSII internal electron acceptors ($Q_A$ and Pheophytin), the plastocyanin downstream of the cytochrome $b_6f$ complex and the PQ connected to PSII (the photochemically active pool). Using this approach, we found that the ETC electron capacity is not different between *pgr6* and wild type when plants were grown under moderate light. This suggests that the photoactive PQ pool has the same electron capacity in *pgr6* and wild type at least under moderate light. On the other hand, we detected a smaller electron capacity in *sps2* mutant, consistent with a constitutive lack of PQ in this line. Upon 3 h of high light exposure, both *pgr6* and *sps2* mutants showed a diminished electron transport capacity (Fig. 3a). This finding suggests that increasing light intensity diminishes the photoactive PQ pool in *pgr6*, thus the phenotype was similar to *sps2* from a functional point of view. To substantiate the hypothesis that the changes in the ETC observed after 3 h of high light are linked to the lack of photoactive PQ, we quantified changes in the electron fluxes in the different genotypes using a previously established kinetic model based on fluorescence induction kinetics (the JIP test)[33,34]. We found that the maximum quantum yield of primary photochemistry in PSII (ΦPo) did not vary between wild type and *pgr6* under moderate light (Supplementary Fig. 4), consistent with the previous measurements (Fig. 1b). However, the quantum yield of the electron transport flux after $Q_A$ (ΦET2o) and the yield of electron transport to PSI electron acceptors (ΦRE1o) were already lower in *pgr6* mutants compared to wild type. After 3 h of high light exposure, the maximum yield was again not affected by the *pgr6* mutation, however both parameters related to the transport from PSII to PSI (ΦET2o and ΦRE1o) were even further decreased in the two *pgr6* mutant lines (ΦET2o: 0.28 ± 0.04; 0.28 ± 0.07) (ΦRE1o: 0.08 ± 0.02; 0.07 ± 0.03) compared to wild type (ΦET2o: 0.38 ± 0.03; ΦRE1o: 0.13 ± 0.02). Similarly, the PQ-limited *sps2* plants showed lower electron transport efficiency (lower ΦET2o and ΦRE1o) under both light conditions (Supplementary Fig. 4). This analysis points to a constitutive lower capacity of the mutants to perform electron transport from PSII to PSI, which is consistent with a perturbation of the photoactive PQ pool in *pgr6*. However, this defect becomes symptomatic only upon exposure to high light. A direct measurement in *pgr6* of the fraction of the PSII reaction centres incapable of transferring electrons to the ETC (closed) by variable room temperature fluorescence upon exposure to increasing light intensities[35] also supports this hypothesis. In fact, the steady-state fluorescence was higher in the mutant confirming that the PSII reaction centres were systematically more closed in *pgr6* than in the wild type (Supplementary Fig. 5). Closure of PSII reaction centres is

the expected consequence of a limitation of the electron transfer to the photoactive PQ pool[35]. It is worth noting that said effect is already measurable in plants not exposed to high light, suggesting that the electron transport efficiency is constitutively defective in *pgr6*.

A limitation in the size of the photoactive PQ pool should also have a downstream effect on PSI, by decreasing the number of electrons available to reduce its primary electron donor ($P_{700}$) upon light-driven oxidation. This hypothesis can be tested by comparing the lag time of $P_{700}^+$ oxidation in conditions under which the ETC is completely oxidised (devoid of transportable electrons so that only the one electron present inside the PSI will account for the delay) or fully reduced by a saturating flash. The ratio between these two values provides the number of electrons contained in the whole ETC per PSI[6]. We measured these parameters using time resolved redox spectroscopy to quantify the *pgr6*-induced defect[6]. Wild-type plants, acclimated to moderate light, had a maximum of 19 ± 2 electrons per PSI, which is a number consistent with previous reports estimating the number of electron carriers[36], whereas in both *pgr6* lines background there were only 8 ± 1 electrons per PSI. After 3 h of high light exposure, the ETC of the wild-type plant contained 13 ± 2 electrons per PSI, while only 4 ± 1 and 3 ± 1 were present in *pgr6-1* and *pgr6-2*, respectively (Fig. 3b and Table 1). These results demonstrate that the ETC capacity is limited in *pgr6* and that this defect is accentuated by the high light treatment. Interestingly, the same effect was observed in the *sps2* background (Table 1).

In summary, the spectroscopic data for both *pgr6* and *sps2* are consistent with a scenario in which the limitation of photoactive PQ results in diminished electron transport. Importantly, also, the cytochrome $b_6f$ turnover rate[37] was affected in neither *pgr6* nor in *sps2*, featuring wild-type kinetics after exposure to 3 h of high light (Supplementary Fig. 6). This indicates that the very high affinity of this complex for PQ in vivo ensures maximum turnover even under conditions when the PQ pool is lowered[38].

To biochemically determine the size of the photoactive PQ pool, the amounts of reduced and oxidised PQ were measured by HPLC in leaves in which the photoactive PQ was either fully oxidised by 2min of FAR light or fully reduced by saturating white light[4,5,11,12]. Total PQ in *pgr6* was indistinguishable from the wild type (Fig. 3c), and the photoactive PQ was 0.33 ± 0.08 nmoles cm⁻² in wild type and 0.24 ± 0.05 and 0.23 ± 0.06 nmoles cm⁻² in *pgr6-1* and *pgr6-2*, respectively, under moderate light (Fig. 3d). However, after 3-h high light, the photoactive PQ in *pgr6* mutants decreased significantly (Student *t* test, $p = 0.05$) to 0.14 ± 0.05 nmoles cm⁻² and 0.14 ±

0.02 nmoles cm$^{-2}$, while the wild-type photoactive PQ pool remained stable around 0.30 ± 0.13 nmoles cm$^{-2}$ (Fig. 3d). Furthermore, no major difference was observed in the levels of the hydroxyl-PQ, a molecule that accumulates when PQ is depleted by oxidative stress, between wild type, pgr6 and sps2[12] (Supplementary Fig. 7). These results demonstrate that high light treatment depletes photoactive PQ in pgr6. Since total PQ was not measurably different under either moderate or high light conditions, no accumulation of the oxidation product hydroxyl-PQ had occurred and the photoactive pool was smaller, we expected the non-photoactive PQ pool in the plastoglobules to be increased. Consistently, higher levels of PQ were present in the plastoglobules isolated from pgr6 and this difference was accentuated after 3-h high light (Supplementary Fig. 8).

## Discussion

Knockout mutants of *PGR6* are characterised by conditional defects in growth and development, including the pale cotyledon phenotype previously reported under constant red light[18] and the variegated phenotype under high light reported here (Fig. 1a). So far, no molecular explanation for the observed defects was provided besides proposing that either the lower photosynthetic efficiency[17] or the misregulation of the prenyl-lipid metabolism[19] in pgr6 are the cause of the biochemical phenotype upon prolonged high light. In this work, we analysed plants grown under moderate non-phenotype-inducing light and assessed their photosynthetic traits after a limited exposure to high light. We observed that 3 h of high light were sufficient to trigger a clear photosynthetic defect in pgr6, suggesting that the underlying cause was either already present before the treatment or could be attributed to the lack of fast adaptive responses. Photosynthetic complexes were affected neither in amount nor in activity, suggesting that the defect is not a direct consequence of protein perturbation (Supplementary Figs. 1 and 6).

A first level of response to photosynthetic imbalance is through the phosphorylation network of the thylakoid proteins controlled, mostly, by the kinases STN7 and STN8 and their counteracting phosphatases[23,30]. Due to the STN7- and STN8-dependent phosphorylation, the photosynthetic apparatus is capable to cope with rapid changes in the environmental conditions by maintaining an optimal photosynthetic efficiency (e.g. state transitions) and cope with damages to the photosystems (e.g. regulation of D1 repair cycle)[39,40]. In high light, the overall thylakoid protein phosphorylation was decreased in pgr6 compared to the wild type. Lower activity of the STN7 kinase is expected in plants shifted to high light, however, the analysis of the phosphorylation pattern indicated lower activities of both STN7 and STN8 kinases in pgr6 (Fig. 2). The activity of both STN kinases is linked to the redox status of the PQ pool. Therefore, this defect can potentially be explained by an influence of PGR6 on PQ. Although we cannot fully exclude that PGR6 is a direct regulator of the STN kinases, this scenario seems a rather unlikely explanation for the photosynthetic defects observed in pgr6. Indeed, even the double knockout mutant of the STN kinases (stn7/stn8), which has an even lower level of phosphorylation of the target proteins than pgr6, does not display a defect in electron transport comparable to that of pgr6 (Supplementary Fig. 2). Furthermore, sps2, which is genetically deprived of PQ, displays a phosphorylation defect similar to pgr6 (Fig. 2).

The evidence so far points toward PQ as a key molecule regulated by PGR6 activity. Biochemical and biophysical analyses were performed to assess whether there is a limitation of the photoactive PQ pool in pgr6 (Fig. 3). The comparison between wild type, pgr6 and sps2 offers a suitable experimental system to test this model. Indeed, when the photoactive PQ pool is genetically limited, as in sps2, the

PSII input will be in excess over the electron capacity of the PQ pool at lower light intensity than in wild type (Supplementary Fig. 3). As a result, the fraction of closed PSII, unable to donate electrons to the photoactive PQ, will increase and thus limit the photosynthetic efficiency compared to wild type.

These data show that here the role of PGR6 is to maintain the size of the photoactive PQ pool, i.e. the PQ available for the photosynthetic ETC. The lack of PGR6 does not result in a visible phenotype under moderate light indicating that its activity is not essential under this light condition[13]. This can be rationalised assuming that the loss of photoactive PQ depends on the electron flux from the PSII. The electron flow is the result of the photon input (i.e. light intensity) minus the portion of absorbed photons dissipated as heat (NPQ)[41]. It is only during high light that the photoactive PQ pool will receive electrons in excess of the electron flow capacity and therefore an efficient supply of PQ from a reservoir (non-photoactive PQ pool) is required for its homoeostasis[12]. In this model, the role of PGR6 is to ensure a rapid refill of the photoactive PQ pool, which is essential in high electron fluencies (high light) (Fig. 4). Consequently, in a pgr6 mutant upon high light exposure, the photoactive PQ pool will be depleted limiting the amount of the electron carriers available for photosynthesis. Hydroxylation of the photoactive PQ may be the cause of said depletion, which becomes phenotypic when combined with an inefficient supply of newly synthesised PQ from storage compartments to the photoactive pool[12]. However, no detectable difference in the level of accumulation of the hydroxyl-PQ has been observed between pgr6 and the wild type (Supplementary Fig. 7). Although the most likely storage compartments capable of efficiently and quickly refilling the photoactive PQ are the plastoglobules, the contribution of the envelope-located PQ cannot be excluded.

The depletion of the photoactive PQ pool observed in pgr6 upon exposure to 3-h high light may account for the reported electron transport limitation and resembles that of sps2. The observation that total PQ is not affected and there is no accumulation of hydroxyl-PQ in pgr6 after 3 h of high light supports the model in which the cause for photoactive PQ pool depletion is the lack of an efficient refill from the PQ reservoir[12]. The lower refill ratio can be explained by a lower mobility of PQ in pgr6. PQ mobility constraints would also explain the measurable increase in the fraction of closed PSII reaction centres (1-qP) and NPQ observed in pgr6 plants grown under moderate light, where the photoactive PQ pool size is unaffected (Figs. 1, 3 and Supplementary Fig. 5) and the lower yield of electron transport from PSII to PQ (ΦET2o) (Supplementary Fig. 4). Additional evidence comes from measuring the electron transport as the output of the ETC at the level of P$_{700}$ (PSI) oxidation. The output defect of pgr6 and sps2 appears to be much larger than the lack of photoactive PQ measured biochemically. This is exceedingly evident in the sps2 mutant, where the depletion of PQ caused by high light resulted in a drop of the measured electron transport to less than 1 electron, suggesting that the ETC is almost completely blocked (Table 1). However, there is still a measurable amount of available electron acceptors for PSII and therefore of available PQ molecules (Fig. 3a). The observation is consistent with previous studies on the sps2 mutant[4] and becomes highly relevant in the context of the defect in pgr6: the photoactive PQ pool appears only slightly diminished (Fig. 3d), but electron transport is disproportionately affected (Figs. 1c and 3a, b). This is consistent with the previous model, showing that a lower concentration of PQ in the photoactive pool cannot efficiently overcome the diffusion barriers affecting its access to the photosynthetic complexes[42]. Furthermore, by limiting its own mobility in the thylakoid membranes[42], the exchange with the non-photoactive PQ pool will be impaired. Therefore, a combination of adequate photoactive PQ pool size,

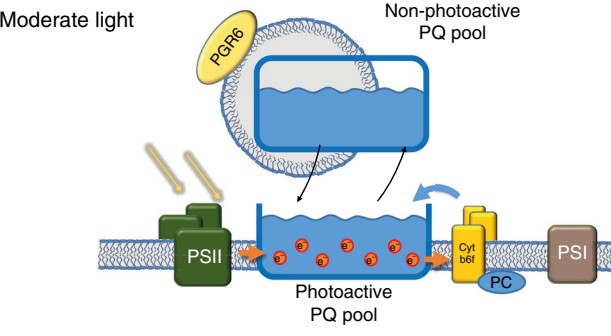

Moderate light

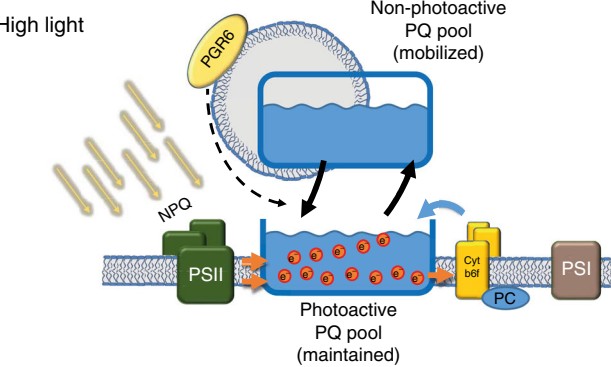

High light

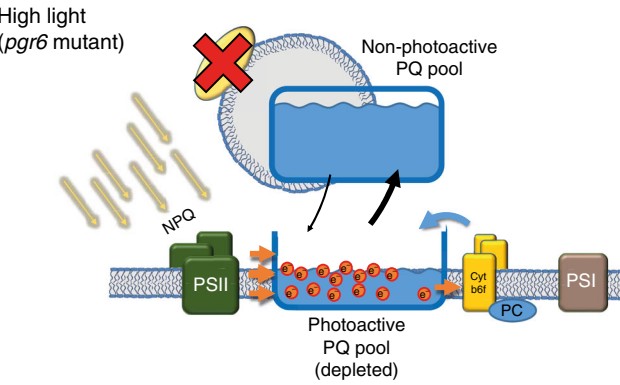

High light (pgr6 mutant)

**Fig. 4** Schematic representation of the regulation mediated by PGR6. Under moderate light intensity, the electron input from the PSII is compensated by the activity of cytochrome $b_6f$ complex. Under this condition, the action of the two complexes is in equilibrium and maintains the photoactive PQ pool in balance thus allowing a continuous electron transport. When the light exceeds the electron transport capacity (high light), the electron input from the PSII is higher than the output from the cytochrome $b_6f$, this effect is partially mitigated by the thermal dissipation of light excess (NPQ). Under high light, the maintenance of the photoactive PQ depends as well on the mobilisation of the reservoir, i.e. the PQ stored in the non-photoactive pool. This mobilisation is possible thanks to the activity of PGR6, which regulates this redistribution thus allowing the preservation of the photoactive PQ pool under high light. In the absence of PGR6, the input from the non-photoactive pool is limited and therefore insufficient to replenish the photoactive PQ pool, furthermore, the lower NPQ induction causes an even stronger overreduction of the PQ pool, which further increases the loss of the photoactive PQ. These combined defects result in a reduction of growth and in developmental issues observed in pgr6 mutant plants. Orange dots represent the electrons contained inside the pool, orange arrows represent the movement of the reduced PQ and blue arrows represent the movement of the oxidised form, while black arrows represent the rate of exchange between the photoactive and the non-photoactive PQ pools

PQ mobility and refill rate are required to ensure PQ homoeostasis and photosynthetic efficiency. Perturbation of the electron flow through the photoactive PQ, which is also the molecular vector for proton transfer across the thylakoid membrane, will limit the trans-thylakoid proton gradient needed to activate the thermal dissipation of excess absorbed light (NPQ)[43]. With a lower NPQ, the ETC will become even more overreduced since the electron input from PSII cannot be adequately controlled. This creates a negative loop that will further exacerbate the electron transport defect. The finding that both pgr6 and sps2 share common defects at the level of electron transport and NPQ (Fig. 1 and Supplementary Fig. 3) is also in accordance with this model. In the long term (i.e. several days of high light), these combined defects increase the high light dependent degradation of PQ[12] and will result in the depletion of the total PQ that was previously reported in sps2[11,12] and pgr6[17].

A defect in PQ mobility and availability may also explain the previously reported defect in carotenoid accumulation in pgr6[17,19]: Oxidised PQ functions as a sink for the electrons removed during the desaturase reactions of carotenoid biosynthesis; a limitation in the access, mobility and/or concentration of PQ may hamper this process and would lead to a lower level of carotenoid production. The consequence of this scenario would be similar to the one of immutans that lacks the terminal PQ oxidase[21]. Interestingly, this mutant is also characterised by a variegated light-dependent phenotype similar to the one observed in pgr6 exposed to high light. Finally, alteration of the size (and accessibility) of the photoactive PQ pool would explain long-term effects of pgr6 on chloroplast to nucleus communication, thus providing a rationale for the pale cotyledon phenotype reported when grown under constant red light[44].

In conclusion, our results indicate that in order to maintain photoactive PQ as well as photosynthetic efficiency while preventing long-term photodamage[17,19] under high light, plants have evolved a PQ homoeostasis mechanism controlled by the PGR6 kinase (Fig. 4). The observed depletion of photoactive PQ in pgr6 after 3-h high light explains the diminished STN7 activity on LHCII together with the measured effects on state transitions. Also, the proton gradient regulation phenotype considering that the lack of photoactive PQ will generate a smaller trans-thylakoid proton gradient, which is needed to activate the NPQ mechanism[43], and the lack of this protective mechanism results in increased chlorophyll fluorescence, which is the signature phenotype of all pgr mutants[13]. Considering its subcellular localisation on plastoglobules, it is tempting to hypothesise that the role of PGR6 is to control the release of PQ from plastoglobules supplying the ETC in the thylakoid membrane[12]. In support of this model, there was a measurable increase in PQ concentration in pgr6 plastoglobules compared to wild type suggesting PQ trapping in the non-photoactive pool (Supplementary Fig. 8).

## Methods

**Plants material and treatments**. *Arabidopsis thaliana* wild-type plant refers to var. Columbia-0 (Col-0). pgr6-1 T-DNA insertion line (SALK_068628) and pgr6-2 T-DNA insertion line (SALK_130499C) were purchased at Nottingham Arabidopsis Stock Centre (NASC, http://arabidopsis.info). stn7/stn8 and sps2 were gifts from Prof. Goldschmidt-Clermont and from Prof. Basset, respectively.

Seedlings were grown for 14 days in 0.5× MS plates under continuous light condition (80μmol m$^{-2}$ s$^{-1}$ for control and 500μmol m$^{-2}$ s$^{-1}$ for the high light). Plants were grown on soil with solbac (Andermatt) under moderate light conditions (120μmol m$^{-2}$ s$^{-1}$, 20–22 °C and 8 h light/16 h dark) in a controlled environment room. For high light treatment, 4–5-week-old plants were exposed to 500μmol m$^{-2}$ s$^{-1}$, 20–22 °C, for 3 h.

Samples were collected under light, directly frozen in liquid nitrogen and stored at −20 °C.

**Photosynthetic parameters**. Maximum quantum yield of PSII ($\Phi_{MAX}$), quantum yield of PSII ($\Phi_{PSII}$) and NPQ were determined using Fluorcam (Photon System Instrument, Czech Republic, http://www.psi.cz) with blue light LED (470nm). Plants were dark adapted for 10min before measurements. $\Phi_{MAX} = (F_V/F_M)$; $\Phi_{PSII} = (F_M'-F_S)/F_M'$; and $NPQ = (F_M-F_M')/F_M'$; where $F_M$ is maximum fluorescence; $F_0$ is minimum fluorescence; $F_V$ is the variable fluorescence ($F_M-F_0$) in dark-adapted state; $F_M'$ is maximum fluorescence; and $F_S$ is steady-state chlorophyll fluorescence in the light[35]. The employed PPFD, (photosynthetic photon flux density), measured by LI-189 photometer (LI-COR), are 2.5–95–347–610–876–1145µmol m$^{-2}$ s$^{-1}$. State transitions were measured with the same instrument. After measurement of the $F_0$ and $F_M$, plants were exposed to 10min red light (50µmol m$^{-2}$ s$^{-1}$ 660nm peak measured as PPFD) supplemented with far red (17µmol m$^{-2}$ s$^{-1}$ calculated from the 733nm peak area considering values between 500 and 800nm). At the end of this phase, the $F_M'$ ($F_M$ST1) was measured, and then the FAR light were turned off. The transition from state 1 to state 2 was followed during 10min, then again the $F_M'$ ($F_M$ST2) was measured. Quenching related to state transition (qT) was calculated as $qT = (F_M ST1 - F_M ST2)/F_M$.

**Chlorophyll a fluorescence curve kinetics (OJIP, JIP test)**. Fast chlorophyll, $a$ fluorescence induction (OJIP, JIP-test) kinetics, were measured at room temperature using a plant efficiency analyser (Handy-PEA; Hansatech Ltd., King's Lynn, Norfolk, England), following manufacturer instructions. Plants were dark adapted for 10min before measurements. Measured data were extracted with the WinPEA software (Hansatech) and analysed with JIP-test according to Strasser et al. (2010)[33] and Kalaji et al. (2014)[34]. In detail, ΦPo (maximum quantum yield of primary PSII photochemistry) was calculated as $1 - F_0/F_M$. ΦET2o (quantum yield of the electron transport from $Q_A$ to $Q_B$) as $((F_M - F_0)/F_M)\,(1 - (F_{2ms} - F_0)/(F_M - F_0))$. ΦRE1o (quantum yield of the electron transport until the PSI electron acceptors) as $((F_M - F_0)/F_M)\,(1 - (F_{30ms} - F_0)/(F_M - F_0))$. Where $F_M$ is the maximum fluorescence, $F_0$ the minimal fluorescence calculated by the Handy-PEA, $F_{2\,ms}$ and $F_{30\,ms}$ are the fluorescence levels measured at 2 and 30ms, respectively.

**$P_{700}$ oxidation**. The kinetics of PSI photoxidation were measured on detached leaves using a JTS-10 LED spectrometer (BioLogic Science Instruments) in absorbance mode.

$P_{700}$ oxidation was assessed by increase in absorption at 810nm (after deconvolution of plastocyanin absorption, as described in Joliot and Joliot[6]). FAR illumination was provided by a LED peaking at 735 nm, filtered through three Wratten filters 55 that block wavelengths shorter than 700nm. When needed, the maximum extent of $P_{700}^+$ was estimated by imposing a white light saturating flash on top of the FAR. A red LED provided actinic light peaking at 640nm[45]. In order to measure the number of electrons present in the ETC per PSI, the plants were incubated 2min under strong white light (500µmol m$^{-2}$ s$^{-1}$) in order to reduce the contribution of the cyclic electron flow by activating $CO_2$ assimilation in the leaves[6]. Reactivation of cyclic flow only occurs after a long period of dark[6,45]. Therefore, after a short dark adaptation, electrons available to $P_{700}^+$ are only reflecting the reduction level of the PSI including the PQ pool. We exposed the leaf to FAR for 2min to oxidise the ETC and, after 2 s of dark adaptation to allow $P_{700}$ reduction, we followed its reoxidation induced by FAR either in the presence or in the absence of a short saturating flash of actinic light (1000µmol m$^{-2}$ s$^{-1}$ for 100 µs) to fully reduce the ETC. The time interval between the beginning of FAR illumination and the beginning of $P_{700}$ oxidation was measured after a saturating pulse (PSI electron donors reduced) and after dark incubation (PSI electron donors oxidised). The ratio between these two values is used as a proxy for the number of available electrons per PSI (Supplementary Fig. 9).

**Immunoblot analysis**. Total proteins were extracted from *Arabidopsis* light-exposed leaves and homogenised in 400µL of lysis buffer (100mM Tris-HCl pH 8.5, 2% SDS, 10mM NaF and 0.05% of protease inhibitor cocktail for plant (Sigma)) with a micro pestle in a 1.5mL microtube. Proteins were denatured at 37 °C for 30minu, then centrifuged for 5 min at 16,000 $g$ at room temperature. Two hundred microliters of supernatant were precipitated by chloroform–methanol, then resuspended in sample buffer (50mM Tris-HCl pH 6.8, 100mM dithiothreitol, 2% SDS, 0.1% bromophenol blue and 10% Glycerol) at 0.5 µg chlorophyll per µL and denatured at 65 °C for 10 min. Five–ten microliters of supernatant was mixed with 1mL of 80% acetone and chlorophylls concentration was determined according to Arnon (1949)[46]. An amount of thylakoids equivalent to 2 µg of chlorophyll were loaded. Proteins were separated by 12% SDS-PAGE and transferred onto a nitrocellulose membrane for western blotting.

For Phostag™-pendant acrylamide gels, we followed the protocol for the antenna proteins previously described in Longoni et al.[47]. For the detection of PSII core subunits and STN7 phopshorylation, the protocol was modified as follows: a Phostag™ gradient (0 to 25µM) and Zn(NO₃)₂ (0 to50 µM) was made in the upper half of the resolving 7% acrylamide in 0.35 M Bis-Tris pH 6.8 gel. The stacking gel (4% acrylamide and 0.35 M Bis-Tris pH 6.8) was casted above the resolving gel. The gels were incubated at room temperature for at least 3 h before loading. Samples were prepared as previously described in Longoni et al.[47]. Briefly, total leaf protein were ground in liquid nitrogen and resuspend in lysis buffer (100mM Tris HCl pH 7.8, 2% SDS, 10 mM NaF, 1× cOmplete™ and EDTA-free protease inhibitor cocktail (Roche)). Following incubation at 37 °C for 30′ in agitation (1000 r.p.m Eppendorf thermomixer) protein content was measured with the Bicinchoninic Acid Protein Assay Kit (Sigma-Aldrich) and samples were diluted to equal protein concentration (0.5µg/µL). Samples were further diluted in 2× lithium dodecyl sulfate (LDS) loading buffer (10% glycerol, 244mM Tris HCl pH 8.5, 2% LDS, 0.33 mM Coomassie Brilliant Blue G-250 and 100mM dithiothreitol) and heated for 5 min at 70 °C before loading.

Immunodetections were performed using anti-Actin (Sigma, A 0480) at 1/3000 dilution in 5% fat free milk/PBS, anti-Lhcb1 (Agrisera, AS09 522), anti-Lhcb2 (Agrisera, AS01 003), anti-D1 (PsbA) (Agrisera, AS05 084), anti-PsbO (Agrisera, AS14 2825), anti-PetC (Agrisera, AS08 330); anti-PsaD (Agrisera, AS09 461), anti-PsaC (Agrisera, AS04 042P), anti-AtpC (Agrisera, AS08 312); anti-STN7 (Agrisera, AS16 4098), anti-PsbD (Agrisera, AS06 146) at 1/5000 dilution in 5% fat free milk/ TBS and anti-Phosphothreonine (Cell Signaling Technology, #9381) at 1/10'000 in 3% BSA/TBS Tween20 0.1%. Secondary antibodies (anti-rabbit (Merck, AP132P) or anti-mouse (Sigma, A5278) at 1/3000 conjugated with HRP allow the detection of proteins of interest with 1mL of enhanced chemiluminescence and 3.3µL of $H_2O_2$ 3% using an imager for chemiluminescence (Amersham Imager 600, Amersham Biosciences, Inc).

**PQ analysis**. Small leaf discs (0.8cm diameter) were taken from 5-week-old plants. Total lipids were extracted after 15 s of saturating white light (2000µmol m$^{-2}$ s$^{-1}$) using a fibre-optic system allowing a maximal reduction of the PQ pool. Samples were directly flash frozen in liquid nitrogen at the end of the light treatment while still illuminated. A second disc from the same leaf was treated with FAR light (735nm and 5.5 µmol m$^{-2}$ s$^{-1}$) for 2 min allowing a maximal oxidation of the PQ pool. For prenyl lipid determination, samples were grinded immediately in the frozen state and extracted with cold ethyl acetate. This step as well as the analyses were performed as described in Kruk and Karpinski[5] and Ksas et al.[11,12]. The photoactive PQ pool was determined from the difference between the reduced PQ after 2 min of FAR light, upon which all the photoactive pool is oxidised, and the reduced PQ after a high irradiance light flash, upon which all the photoactive pool is reduced. Plastoglobule isolation from chloroplasts and prenyl lipid analysis were performed according to Martinis et al.[48] and Eugeni-Piller et al.[49]. Intact chloroplast were extracted from entire leaves by grinding in HB buffer (Sorbitol 450mM, Tricine-KOH pH 8.4 20mM, EDTA pH 8.4 10mM, NaHCO₃ 10mM, MnCl₂ 1mM, Na-ascorbate 5mM and PMSF 1mM). The chloroplast were filtered through two layers of miracloth (Merck Millipore) and collected by centrifugation (5600 × $g$). The chloroplasts were lysate by osmotic shock in TED buffer (Tricine pH 7.5 50mM, EDTA-Na₂ 2mM and dithiothreitol 2mM) supplemented with 0.6 M sucrose. Chloroplasts were diluted to a concentration of 2 mg/ml of chlorophyll and incubated for 10′ on ice to allow a complete lysis and further incubated for 2 h at −80 °C. The samples were diluted four times in TED buffer. The sample was homogenate 20 times with a Dounce homogeniser (PTFE Tissue grinder 50cm3, VWR®). The membranes and plastoglobules were separated from the stroma by ultracentrifugation (60′ 100,000 × $g$ at 4 °C). The pellet was dissolved in TED buffer supplemented with 45% sucrose to a concentration of 2–3mg/mL of chlorophyll. Further homogenisation of the sample was performed with a Dounce homogeniser (20 times). This solution was used as lower phase of a discontinuous sucrose gradient. The gradient was assembled in TED buffer with the following sucrose concentrations: 15ml of sample in 45% sucrose, 6mL of 38% sucrose, 6mL of 20% sucrose, 4mL of 15% sucrose and 8mL of 5% sucrose. The gradient was centrifuged to allow the fractionation by flotation (16 h, 100,000 × $g$ at 4 °C). One-milliliter fractions were collected from the top of the gradient. The lipids from each fraction were extracted with ethyl -acetate (0.75 volumes, 2 times), the ethyl-acetate phase was recovered upon centrifugation (1′ 10,000 × $g$) and dried in a speedvac. The dried pellet was solubilized in a tetrahydrofuran–methanol (1:1) solution, and used for UHPLC-APCI-MS-QTOF analysis.

**Statistics and reproducibility**. The sample size was determined empirically for each experiment (minimum of three independent organism and two experimental replicates), on the basis of experience with similar assays and from sample sizes generally used by other investigators. No data were excluded from the analysis. The experiment was replicated at least two times, the results were reproducible when the plants were not stressed before the experiment. When testing light condition, the position of the plants of different genotypes was changed randomly in order to reduce any possible positioning effect. The data were compared for statistical difference by a two-tailed, heteroscedastic Student's $t$ test (Excel 2016).

**Reporting Summary**. Further information on research design is available in the Nature Research Reporting Summary linked to this article.

## Data availability

The datasets analysed in this paper are included in this published article (and its supplementary information files). Further datasets generated during the current study are available from the corresponding author on reasonable request.

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

## Acknowledgements

This work was supported by the Swiss National Science Foundation (SNSF) grants 31003A_156998 and 31003A_176191. G.F. acknowledges funding by the HFSP RGP0052 project and the INRA AAP project PURIST funds form the GRAL (ANR-10-LABX-49-01) labex. We thank Prof. Goldschmidt-Clermont for his helpful discussion and support during the project.

## Author contributions

T.P., V.S., P.L. and F.K. designed experiments. T.P., P.L., G.G., B.K., J.C., S.D., M.H. and G.F. performed all experiments. T.P., V.S., P.L., M.H., G.F. and F.K. contributed to the

analysis and the interpretation of the results. T.P., V.S., P.L., M.H., G.F. and F.K. wrote the manuscript.

## Additional information

**Competing interests:** The authors declare no competing interests.

