## [Peer Review File · Communications Biology]

Reviewers' comments:

Reviewer #1 (Remarks to the Author):

Review of Palon T, Shanmugabalaji V, Longoni P, Glauser G, Ksas B, Collombat J, Desmeules S, Havaux M, Finazzi G, Kessler F: Plastoquinone homeostasis by Proton Gradient Regulation 6 is essential for photosynthetic efficiency.

Submitted to Communications Biology 2019

General judgment

The manuscript describes experiments characterizing the role of the PGR6 protein, on the basis of the analysis of two T-DNA inactivation strains. The mutant lines function similarly, strongly suggesting that the differences to the wild type are caused by the inactivation of the PGR6 gene. The inactivation mutants have a slightly lower yield of PSII electron transfer, and this yield is further lowered by 3-h treatment in high light. However, no significant differences in photosystems between the mutant and the wild type are found. The mutant has less phosphorylation of both LHCII and the PSII core proteins and is, especially after a high-light-treatment, defective in state transitions. Another mutant, known to be defective in plastoquinone synthesis, has a similar phenotype. The photoactive plastoquinone pool decreases in the mutant strongly during a 3-h high-light treatment, although the total amount of plastoquinone does not decrease. The authors suggest that the PGR6 protein is required to maintain the photoactive plastoquinone pool through a flow of plastoquinone molecules from plastoglobuli to thylakoids.

The manuscript is very well written, and the conclusions are well supported by the data. I have a few comments that should be taken into account, but they actually do not challenge the main conclusion.

Major comments

1. The electron transfer rate (ETR) through PSII is the product of PPF, quantum yield of PSII electron transfer, leaf absorptance and the fraction of absorbed light absorbed by PSII. Very often, leaf absorptance and fraction of light absorbed by PSII are not measured but assumptions are given. This is fine, as long as relative values are compared, and one can be sure that the two un-measured quantities are the same in all samples. Such an assumption cannot be done when a photosynthetic mutant is compared to the wild type, or a mutant to mutant. Both quantities can be different. Therefore, if the authors wish to keep the ETR values, then the two quantities, or reliable proxies for them, must be measured.

Furthermore, the formula given on line 325 forgets that an Arabidopsis would probably not absorb 100 % of light, even if the wavelength is 470 nm. If the message is that the absorptance at this wavelength in all studied leaves is essentially 100 %, this must be backed up with experiments.

2. Lines 393-394. The details of the flash freezing treatment should be given. How fast was the freezing,

and was a cold clamp used? How was it ensured that the redox state of plastoquinone did not change during melting?

3. Plastoquinone is found in the photoactive pool, in plastoglobuli and in the envelope. The authors appear to know about the envelope, as they mention that photoactive plastoquinone is “largely” in plastoglobuli. However, in the discussion concerning the total plastoquinone in a mutant which has changes in the size of the photoactive pool, the envelope-located plastoquinone cannot be forgotten.

Details

Lines 37-39: Earlier data about the disappearance of plastoquinone under ultraviolet light would be important to mention here (see e.g. Trebst and Pistorius 1965 Z Naturforsch 20 and Giacometti et al. 1996 Eur J Biochem 242).

Line 92: something missing

Lines 98-99: The regulation of STN7 is a central topic of the study, and therefore I feel that citations cannot be limited to a review.

Line 109: non in WT

Line 116 and throughout: Please use the traditional names of the D1 and D2 proteins.

Line 117 lower phosphorylation level of what?

Lines 118-120: A third possibility is that the kinases have overlap in target proteins.

Line 112 remove “the” before PSII

Line 150: you invented the excellent term “electron capacity”. Please use it consistently.

Line 172: “Closure of PSII reaction centers is...” (not “This increase is...”).

Lines 172-173: High light causes additional decrease in the size of the plastoquinone pool but no difference can be seen in Figure S5 between ML and HL treated plants. This apparent discrepancy must be discussed.

Lines 194-195: “pool” missing; “amounts of...were”

Lines 224-226: Rephrase.

Line 226 Activities

Line 235 remove comma

Lines 243-245: It is not clear how the suggestion is connected to the more closed reaction centers in the mutants.

Line 330: For 400-700 nm photosynthetically active light, micromoles would be understood without the explanation that they are PPF (well, I still recommend always mentioning PPF). However, for wavelengths outside of this window, micromoles make no sense unless the wavelength range of the measurement also is given.

Line 338: Please describe properly, including length of dark incubation before measurement, PPF of the measurement, and shortly explain how the three parameters used were measured.

Line 358: electron donors

Line 365: rephrase

Line 367: Please go through all abbreviations and check that each one has been used several times.

Line 370a: actually you had two micrograms of thylakoid suspension and loaded an amount equivalent to two micrograms of chlorophyll

Line 377 remove “polymerizing”

Line 379: it is “fat free”, not “free fat”

Line 395 lipid (not lipids)

Line 396: Please go through all references. There is no Kruk et al. 2006 in the reference list.

Line 398 Plastoglobule

Figure 2: Please separate the different curves by a few seconds so that each can be seen (now all curves are on top of each other at the saturating flashes)

Line 591 as a loading... also: levels...were

Lines 592 and 599 the first comma should be a period.

Lines 594-595: add space before nm

Line 598 and throughout: Student’s (not student’s...Student was a pseudonym of the inventor of the t-distribution but he was a professional chemist and not a “student”.)

Line 633 the absence

Line 636 result

Line 650: I do not understand how two different statistical tests are used to get one p value.

Furthermore, what were the factors in the ANOVA? Identity is not a numerical quantity.

Reviewer #2 (Remarks to the Author):

In this contribution Pralon and colleague’s follow-up on the elucidation of the molecular nature of the mechanism of action of the regulatory kinase PGR6. Previous studies had shown that *Arabidopsis thaliana* mutants, in which PGR6 had been deleted, had become sensitive to short (i.e. a few hours) treatment with high intensities of actinic light, because of an impairment of photosynthetic electron transfer (see e.g. refs 15, 17 of the current manuscript).

The current study uses a wide range of measurements (i.e. Western blotting, absorption- and fluorescence spectroscopy and HPLC-based quinone analyses) to resolve this question. The main conclusion from the current study is that the size of the photoactive PQ pool in the thylakoid membrane is significantly decreased in the PGR6 deletion mutants and that hence PGR6 must have a function in the transfer of PQ from the Plastoglobules to the thylakoid membrane.

My evaluation of this manuscript is that it is written in a rather complex style, and that it contains many rather speculative statements. The methodology applied is well-established, be it that some details (like the method to homogenize the leaves) are lacking. Statistics are routinely applied, but I doubt whether all conclusions on this aspect are warranted (like the differences between the strains in Figs S6 and S7, and why is Φ_{RE10} of *pgr6-1* in Table S4 not in bold? - but I am not an expert in statistics. Also, it would help to support the interpretation of the Western experiments with quantitative (densitometric) analyses.

The experiments displayed in Fig 1a,b can be deleted in view of what is known already from the

literature. Furthermore, in the analyses of PQ content of the thylakoid membrane, it would be worthwhile to include quantitation of hydroxyl-plastoquinone/nol. And in the Discussion it would help to discuss the redox state of the PQ in the Plastoglobules.

Dear Editor,

We addressed all the issues raised by the two reviewers and we resubmit a new version of the manuscript with all the changes marked in red. We are grateful for the comments and we do believe that they really helped us improving the final manuscript.

Please find the answers to each comment below.

Best regards,

Felix Kessler
Paolo Longoni

Reviewers' comments:

Reviewer #1 (Remarks to the Author):

Review of Pralon T, Shanmugabalaji V, Longoni P, Glauser G, Ksas B, Collombat J, Desmeules S, Havaux M, Finazzi G, Kessler F: Plastoquinone homeostasis by Proton Gradient Regulation 6 is essential for photosynthetic efficiency.

Submitted to Communications Biology 2019

General judgment

The manuscript describes experiments characterizing the role of the PGR6 protein, on the basis of the analysis of two T-DNA inactivation strains. The mutant lines function similarly, strongly suggesting that the differences to the wild type are caused by the inactivation of the PGR6 gene. The inactivation mutants have a slightly lower yield of PSII electron transfer, and this yield is further lowered by 3-h treatment in high light. However, no significant differences in photosystems between the mutant and the wild type are found. The mutant has less phosphorylation of both LHCI and the PSII core proteins and is, especially after a high-light-treatment, defective in state transitions. Another mutant, known to be defective in plastoquinone synthesis, has a similar phenotype. The photoactive plastoquinone pool decreases in the mutant strongly during a 3-h high-light treatment, although the total amount of plastoquinone does not decrease. The authors suggest that the PGR6 protein is required to maintain the photoactive plastoquinone pool through a flow of plastoquinone molecules from plastoglobuli to thylakoids.

The manuscript is very well written, and the conclusions are well supported by the data. I have a few comments that should be taken into account, but they actually do not challenge the main conclusion.

Major comments

1. The electron transfer rate (ETR) through PSII is the product of PPFD, quantum yield of PSII electron transfer, leaf absorbance and the fraction of absorbed light absorbed by PSII. Very often, leaf absorbance and fraction of light absorbed by PSII are not measured but assumptions are given. This is fine, as long as relative values are compared, and one can be sure that the two un-measured quantities are the same in all samples. Such an assumption cannot be done when a photosynthetic mutant is compared to the wild type, or a mutant to mutant. Both quantities can be different. Therefore, if the authors wish to keep the ETR values, then the two quantities, or reliable proxies for them, must be measured.

Furthermore, the formula given on line 325 forgets that an Arabidopsis would probably not absorb 100 % of light, even if the wavelength is 470 nm. If the message is that the absorbance at this wavelength in all studied leaves is essentially 100 %, this must be backed up with experiments.

We agree that the relative ETR value may be prone to errors and we replace it with PSII quantum yield data to avoid these issues. The text and the figure captions were modified accordingly where needed.

2. Lines 393-394. The details of the flash freezing treatment should be given. How fast was the freezing, and was a cold clamp used? How was it ensured that the redox state of plastoquinone did not change during melting?

The freezing was instantaneous as the small volume samples were flash frozen in liquid nitrogen. We now state that “For prenyl lipid determination samples were grinded immediately in the frozen state and extracted with cold ethyl acetate.” We have earlier established that the plastoquinone redox state does not change using this procedure although we did not use a cold clamp. Details regarding the technique were added to the Materials and Methods section. (lines 406-413)

3. Plastoquinone is found in the photoactive pool, in plastoglobuli and in the envelope. The authors appear to know about the envelope, as they mention that photoactive plastoquinone is “largely” in plastoglobuli. However, in the discussion concerning the total plastoquinone in a mutant which has changes in the size of the photoactive pool, the envelope-located plastoquinone cannot be forgotten.

This is a very valid point and we now added the following sentence to the discussion: “Although the most likely storage compartments capable of efficiently and quickly refilling the photoactive PQ are the plastoglobules, the contribution of the envelope-located PQ cannot be excluded.” (lines 256-260)

Details

Lines 37-39: Earlier data about the disappearance of plastoquinone under ultraviolet light would be important to mention here (see e.g. Trebst and Pistorius 1965 Z Naturforsch 20 and Giacometti et al. 1996 Eur J Biochem 242).

We now refer to these papers. However, we would like to mention here that during the 3h high light treatment the total plastoquinone concentrations were not diminished and that hydroxyl plastoquinone accumulated to the same low levels in all genotypes. Also, see our reply to reviewer 2 regarding the hydroxyl plastoquinone (line 37).

Line 92: something missing

That’s right: we added “compared to WT” in the sentence (line 89)

“... that phosphorylation of both LHCII and PSII was clearly lower in both *pgr6* lines after 3h HL compared to WT (Fig. 2a), while there was no visible difference under moderate light.”

Lines 98-99: The regulation of STN7 is a central topic of the study, and therefore I feel that citations cannot be limited to a review.

Agree: we added primary references for STN7 regulation (Shapiguzov, 2016; Trotta, 2016) (line 96)

Line 109: non in WT

Thanks: “not WT” was changed to “not in WT” (lines 107-108)

Line 116 and throughout: Please use the traditional names of the D1 and D2 proteins.
Comment accepted: we added D1 and D2 to the PsbA and PsbD designation.

Line 117 lower phosphorylation level of what?
Thanks: we added “of D1 (PsbA) and D2 (PsbD)” (line 114)

Lines 118-120: A third possibility is that the kinases have overlap in target proteins.
**That’s certainly a possibility:
We added “... or that they have overlapping target proteins. A decrease in PSII phosphorylation may affect the repair cycle of the core protein D1 (PsbA) thereby decreasing the maximum efficiency of PSII”
(Lines 118-120)**

Line 122 remove “the” before PSII
It is done! Thank you

Line 150: you invented the excellent term “electron capacity”. Please use it consistently.
OK: This suggests that the photoactive PQ pool has the same electron capacity in *pgr6* and WT at least under moderate light. On the other hand, we detected a smaller electron capacity in *sps2* mutant, consistent with a constitutive lack of PQ in this line. (lines 145 and following)

Line 172: “Closure of PSII reaction centers is...” (not “This increase is...”).
Thank you! We changed this (line 169)

Lines 172-173: High light causes additional decrease in the size of the plastoquinone pool but no difference can be seen in Figure S5 between ML and HL treated plants. This apparent discrepancy must be discussed.
Thank you for the observation. We discussed it in the results and in the discussion:

It is worth noting that said effect is already measurable in plants not exposed to high light, suggesting that the electron transport efficiency is constitutively defective in *pgr6*. (line 171)

. The lower refill ratio can be explained by a lower mobility of PQ in *pgr6*. PQ mobility constraints would also explain the measurable limitation in the PSII photochemical efficiency (1-qP) and NPQ observed in *pgr6* plants grown under moderate light, where the photoactive PQ pool size is unaffected (Fig. 1, 3, Supplementary data Fig. 5) and the lower yield of electron transport from PSII to PQ ($\Phi ET2o$) (Supplementary data Fig. 4). (lines 265 - 268)

Lines 194-195: “pool” missing; “amounts of...were”
Thanks: did it (lines 192-193)

Lines 224-226: Rephrase.
**We propose the following formulation:
“Due to the STN7 and STN8-dependent phosphorylation, the photosynthetic apparatus is capable to cope with rapid changes in the environmental conditions by maintaining an optimal photosynthetic efficiency (e.g. state transitions) and cope with damages to the photosystems (e.g. regulation of D1 repair cycle)” (lines 223-226)**

Line 226 Activities
Changed (line 228)

Line 235 remove comma

Thank you, we removed it.

Lines 243-245: It is not clear how the suggestion is connected to the more closed reaction centers in the mutants.

We now say:

“As a result, the fraction of “closed” PSII, unable to donate electrons to the photoactive PQ, will increase and thus limit the photosynthetic efficiency compared to WT.” (lines 242-243)

Line 330: For 400-700 nm photosynthetically active light, micromoles would be understood without the explanation that they are PPF (well, I still recommend always mentioning PPF). However, for wavelengths outside of this window, micromoles make no sense unless the wavelength range of the measurement also is given.

OK, we now say:

“red light ($50 \mu\text{mol}\cdot\text{m}^{-2}\cdot\text{s}^{-1}$ 660 nm peak measured as PPF) supplemented with far-red ($17 \mu\text{mol}\cdot\text{m}^{-2}\cdot\text{s}^{-1}$ calculated from the 733 nm peak area considering values between 500 and 800 nm).” (lines 336-338)

Line 338: Please describe properly, including length of dark incubation before measurement, PPF of the measurement, and shortly explain how the three parameters used were measured.

Thank you. We extended the method descriptions:

$\Phi_{\text{MAX}} = (F_V/F_M)$; $\Phi_{\text{PSII}} = (F_M' - F_S)/F_M'$; $\text{NPQ} = (F_M - F_M')/F_M'$; where F_M , maximum fluorescence; F_0 , minimum fluorescence; F_V the variable fluorescence ($F_M - F_0$) in dark-adapted state; ³⁵. F_M' , maximum fluorescence; F_S , steady-state chlorophyll fluorescence in the light. The employed PPF, (photosynthetic photon flux density), measured by LI-189 photometer (LI-COR), are 2.5 – 95 – 347 – 610 – 876 – 1145 $\mu\text{mol}\cdot\text{m}^{-2}\cdot\text{s}^{-1}$. (lines 331-335)

Plants were dark-adapted for 10 minutes before measurements. (line 345)

In detail, Φ_{Po} (maximum quantum yield of primary PSII photochemistry) was calculated as $1 - F_0/F_M$. Φ_{ET20} (quantum yield of the electron transport from QA to QB) as $((F_M - F_0)/F_M) \cdot (1 - (F_{2\text{ms}} - F_0)/(F_M - F_0))$. Φ_{RE10} (quantum yield of the electron transport until the PSI electron acceptors) as $((F_M - F_0)/F_M) \cdot (1 - (F_{30\text{ms}} - F_0)/(F_M - F_0))$. Where F_M is the maximum fluorescence, F_0 the minimal fluorescence calculated by the Handy-PEA, $F_{2\text{ms}}$ and $F_{30\text{ms}}$ are the fluorescence levels measured at 2 and 30 ms respectively. (lines 347-352)

Line 358: electron donors

Thank you: corrected it

Line 365: rephrase

The sentence was rephrased as follows:

“The time interval between the beginning of FAR illumination and the beginning of P_{700} oxidation was measured after a saturating pulse (PSI electron donors reduced) and after dark incubation (PSI electron donors oxidised).” (lines 368-370)

Line 367: Please go through all abbreviations and check that each one has been used several times.
Did so and removed (DTT) abbreviation.

Line 370a: actually you had two micrograms of thylakoid suspension and loaded an amount equivalent to two micrograms of chlorophyll

We rephrased as follows: “An amount of thylakoids equivalent to two µg of chlorophyll were loaded.” (lines 381-382)

Line 377 remove “polymerizing”

Did so

Line 379: it is “fat free”, not “free fat”

Has been changed in all instances. Thank you

Line 395 lipid (not lipids)

Corrected (line 412)

Line 396: Please go through all references. There is no Kruk et al. 2006 in the reference list.

We checked the references, Kruk et al. 2006 is the reference n°5

Line 398 Plastoglobule

“Plastoglobule isolation” instead of “Plastoglobules isolation” (line 412)

Figure 2: Please separate the different curves by a few seconds so that each can be seen (now all curves are on top of each other at the saturating flashes)

We agree and modified the figure accordingly:

“The fluorescence curves from *pgr6* and *sps2* are shifted on the x axis to allow visualizing the FMST1 and FMST2 values. The x axis time scale refers to the WT curve.” (lines 632-634)

Line 591 as a loading... also: levels...were

Thank you has been corrected: “As a loading...”

“Lhcb1 and Lhcb2 phosphorylation levels were visualized after separation” (line 628).

Lines 592 and 599 the first comma should be a period.

Did so: “acrylamide gels. The upper” (line 629)

Lines 594-595: add space before nm

Did so (lines 631-632)

Line 598 and throughout: Student’s (not student’s...Student was a pseudonym of the inventor of the t-distribution but he was a professional chemist and not a “student”.)

Corrected in all the text (line 638 and throughout)

Line 633 the absence

Corrected (line 672)

Line 636 result

Corrected (line 675)

Line 650: I do not understand how two different statistical tests are used to get one p value. Furthermore, what were the factors in the ANOVA? Identity is not a numerical quantity.

The ANOVA test was performed on the two variables “Moderate Light and High Light”, and the genotypes as groups for each parameter. The ANOVA test was followed by a Student’s t-test from which we extrapolated the statistically different groups, since the letters derive only from the second test we simplified the legend accordingly. (line 687)

Reviewer #2 (Remarks to the Author):

In this contribution Pralon and colleague's follow-up on the elucidation of the molecular nature of the mechanism of action of the regulatory kinase PGR6. Previous studies had shown that *Arabidopsis thaliana* mutants, in which PGR6 had been deleted, had become sensitive to short (i.e. a few hours) treatment with high intensities of actinic light, because of an impairment of photosynthetic electron transfer (see e.g. refs 15, 17 of the current manuscript).

The current study uses a wide range of measurements (i.e. Western blotting, absorption- and fluorescence spectroscopy and HPLC-based quinone analyses) to resolve this question. The main conclusion from the current study is that the size of the photoactive PQ pool in the thylakoid membrane is significantly decreased in the PGR6 deletion mutants and that hence PGR6 must have a function in the transfer of PQ from the Plastoglobules to the thylakoid membrane.

My evaluation of this manuscript is that it is written in a rather complex style, and that it contains many rather speculative statements.

The methodology applied is well-established, but it that some details (like the method to homogenize the leaves) are lacking.

Leaves were homogenized in a microtube with a micro-pestle for protein extraction. Details for homogenization of leaves for photoactive plastoquinone pool were added in the methods section together with the detailed formula to calculate the photoactive pool (lines 374 and following)

Statistics are routinely applied, but I doubt whether all conclusions on this aspect are warranted (like the differences between the strains in Figs S6 and S7, and why is $\Phi RE10$ of *pgr6-1* in Table S4 not in bold? - but I am not an expert in statistics.

Fig. S6 We performed a t-test between the genotypes and reported the p-values.

Fig. S7 (now Suppl. Fig. 8) considering the fact that for the preparation we used a large trail of plants for each data point, the observed values are to be an average of a plant population. In different biological replicates we always observed a clear tendency for *pgr6* to have an higher PQ level in plastoglobules, unfortunately the absolute values are a little bit more noisy so that the difference which is clear in each experimental set becomes less evident when all the points are considered but still significant.

For $\Phi RE10$, despite the difference observed compared to wild type, the p-value of a Student's t-test was higher than 0.05. Since, we agree with the reviewer that this statistical analysis is not completely representative of the true variation we did more measurements and included two additional replicates per genotype and re-calculated the averages so that we could apply the statistical test again. The new results are displayed in the updated table and, including more samples, the difference between *pgr6-1* and WT is significant with a p=0.05 threshold.

Also, it would help to support the interpretation of the Western experiments with quantitative (densitometric) analyses.

That's a good thing: we carried out additional western blots for WT and the *pgr6* lines that allowed densitometric analysis. Critically, they show that after 3h of high light treatment there is no difference in representative components of the photosynthetic complexes. This confirms that no major damage was detectable at the protein level after the short high light treatment. The result of the quantification is presented in Supplementary Figure 1b.

The experiments displayed in Fig 1a,b can be deleted in view of what is known already from the literature.

Fig 1a. The variegated phenotype in young *pgr6* seedlings under constant high light has not been reported so far. We therefore propose not to delete Fig. 1a. This phenotype is different from the tissue necrosis previously reported in adult plants (e.g. Martinis et al. 2014).

Fig 1b. This result has not been reported so far. In the literature (Martinis, Lundquist) the measurements were taken minimum after 24h, while in this paper we measured after only 3h of high light. We show that there is no difference in the QY_{max} after 3h which is an important conclusion justifying the further experimentation. We would therefore propose to leave Fig. 1b in.

Furthermore, in the analyses of PQ content of the thylakoid membrane, it would be worthwhile to include quantitation of hydroxyl-plastoquinone/nol.

This is a good point: We now show the hydroxyl-plastoquinone (Supplementary data Fig. 7). There are no significant differences between WT and *pgr6*, which is in agreement with the overall concentration of PQ remaining the same after 3h high light. (lines 199-201 and 255-260)

And in the Discussion it would help to discuss the redox state of the PQ in the Plastoglobules.

Indeed, this is a good remark. Due to the lengthy plastoglobule purification procedure the initial redox state of PQ will not be preserved. But we know from our earlier work that the overall redox state of PQ in *pgr6* and WT are identical whereas in a mutant that is affected in the plastoglobule PQ redox state it is much more oxidized (Eugeni-Piller et al. PNAS 2014). We therefore assume that PQ redox state in PG is not changed in *pgr6*.

REVIEWERS' COMMENTS:

Reviewer #1 (Remarks to the Author):

The authors have answered to most of my comments. I still have two minor points:

1. Lines 265-286: $1-qP$ is not "PSII photochemical efficiency". $1-qP$ is a measure of the fraction of closed PSII centres. Please correct.
2. Line 296: Reference no 5 is Kruk and Karpinski 2006, not Kruk et al. 2006. Please go through the text, checking for the standard notation for references. The abbreviation "et al." cannot be used to point to one author out of two because it is a plural form (abbreviation from either *et alia*, *et alii* or *et aliae*, which all mean "and others").

REVIEWERS' COMMENTS:

Reviewer #1 (Remarks to the Author):

The authors have answered to most of my comments. I still have two minor points: 1. Lines 265-286: 1-qP is not "PSII photochemical efficiency". 1-qP is a measure of the fraction of closed PSII centres. Please correct.

Corrected as follows: PQ mobility constraints would also explain the measurable increase in the fraction of closed PSII reaction centres (1-qP)

2. Line 296: Reference n:o 5 is Kruk and Karpinski 2006, not Kruk et al. 2006. Please go through the text, checking for the standard notation for references. The abbreviation "et al." cannot be used to point to one author out of two because it is a plural form (abbreviation from either et alia, et alii or et aliae, which all mean "and others").

Corrected